# Rapid detection of methicillin-resistant *Staphylococcus aureus* in positive blood-cultures by recombinase polymerase amplification combined with lateral flow strip

Arpasiri Srisrattakarn[1], Pimchanok Panpru[1], Patcharaporn Tippayawat[1,2], Aroonwadee Chanawong[1,2], Ratree Tavichakorntrakool[1,2], Jureerut Daduang[1,2], Lumyai Wonglakorn[3], Aroonlug Lulitanond [1,2]*

1 Centre for Research and Development of Medical Diagnostic Laboratories, Faculty of Associated Medical Sciences, Khon Kaen University, Khon Kaen, Thailand, 2 Department of Medical Technology, Faculty of Associated Medical Sciences, Khon Kaen University, Khon Kaen, Thailand, 3 Clinical Microbiology Unit, Srinagarind Hospital, Khon Kaen University, Khon Kaen, Thailand

* arolul@kku.ac.th

## Abstract

*Staphylococcus aureus*, especially methicillin-resistant *S. aureus* (MRSA), is an important bacterium that causes community and healthcare-related infections throughout the world. However, the current conventional detection methods are time-consuming. We therefore developed and evaluated a recombinase polymerase amplification-lateral flow strip (RPA-LF) approach for detection of MRSA in positive blood-culture samples. Sixty positive blood-cultures from a hospital were tested directly without DNA extraction and purification before the amplification reaction. RPA primers and probes were designed for *nuc* (encoding thermonuclease) and *mecA* (encoding penicillin-binding protein 2a) genes to diagnose *S. aureus* and its methicillin-resistance status. The RPA reaction occurred under isothermal conditions (45˚C) within 20 min and a result was provided by the LF strip in a further 5 min at room temperature. The evaluation of RPA-LF using blood-culture samples showed 93.3% (14/15) sensitivity for identifying *S. aureus*, and no cross-amplification was seen [100% (45/45) specificity]. For detection of methicillin resistance, the RPA-LF test provided 100% (16/16) sensitivity and 97.7% (43/44) specificity. The RPA-LF is rapid, highly sensitive, robust and easy to use. It can be used for direct detection of MRSA with no requirement for special equipment.

## Introduction

*Staphylococcus aureus* is a Gram-positive bacterium that causes a wide variety of diseases. Its pathogenic potential ranges from minor to severe, the latter category including bloodstream infection and sepsis [1]. Several virulence factors of *S. aureus* cause inflammation and impair immune cell function, thus contributing to serious pathogenesis, which increases the risk of death [2]. *S. aureus* also produces an extracellular thermonuclease, encoded by the *nuc* gene,

**Data Availability Statement:** All relevant data are within the manuscript and its Supporting Information files.

**Funding:** This research was supported by the Fundamental Fund of Khon Kaen University and KKU Research Grant (Project No. I62-00-19-03). The funders had no role in study design, data collection and analysis, decision to publish, or preparation of the manuscript.

**Competing interests:** The authors have declared that no competing interests exist.

presence of which is commonly used to distinguish *S. aureus* from the other *Staphylococcus* spp. [3].

Methicillin-resistant *S. aureus* (MRSA) is now highly prevalent globally and is the biggest threat among Gram-positive pathogens [4]. The resistance depends on the production of a new penicillin-binding protein (PBP2a or PBP2′) encoded by the *mecA* gene. Patients infected with MRSA have a mortality rate about three times higher than those with methicillin-susceptible *S. aureus* [5–7]. Therefore, rapid and accurate detection methods for MRSA in bacteremia are essential for clinical diagnosis to facilitate a specific antimicrobial therapy and reduce the risk of mortality.

The reference standard methods for the detection of MRSA in the blood are traditional culture-biochemical and susceptibility-testing methods. A positive result means bacteria or fungi are present in the blood (positive blood-culture). A negative result means that no signs of any bacteria or fungi were found in the blood. However, they are time-consuming (48–72 h) and laborious [6]. Nucleic-acid amplification tests, such as PCR-based methods, aim for rapid and accurate detection of MRSA, thus avoiding the drawbacks of conventional methods [7]. However, they need special equipment and trained personnel, making them difficult and impractical as point-of-care methods [6, 8].

Recombinase polymerase amplification (RPA) is an isothermal method for DNA amplification that is a promising alternative to PCR. It relies on the actions of three core proteins: recombinase, single-stranded DNA-binding (SSB) protein, and strand-displacing polymerase. The recombinase facilitates the binding of primers with DNA complementary sequences. Polymerase initiates DNA synthesis, whereas the SSB protein stabilizes the DNA strand to prevent primer displacement [9]. RPA is highly sensitive, rapid, and does not require a thermal cycler. It can amplify the target gene across a temperature range from 25 to 45°C within 3–20 min [10]. Moreover, the RPA reaction is also likely to be more robust to the presence of inhibitors than is the PCR method [11, 12]. Currently, amplicons generated by RPA can be detected using agarose gel electrophoresis (AGE), real-time fluorescence and lateral flow (LF) strip. However, the AGE and real-time fluorescence methods need expensive devices and are inconvenient in a hospital setting [13–15]. Detection of RPA products using LF devices is an approach increasingly used for detection of infectious pathogens because it is simple, rapid and readable by the naked eye [16, 17].

Clinical samples often contain both coagulase-negative staphylococci and *S. aureus*, all of which can carry the *mecA* gene [7]. Therefore, detection of *mecA* alone is insufficient for therapeutic decisions: samples should be tested for *S. aureus* in parallel with detection of the *mecA* gene. We previously established an RPA-LF assay for detection of the *mecA* gene (mecA-RPA-LF) [18]. In the present study, we developed an additional RPA-LF assay for *nuc* gene detection (nuc-RPA-LF) and evaluated the performance of both assays (nuc-RPA-LF and mecA-RPA-LF) for the detection of MRSA directly in positive blood-cultures from the hospital.

## Materials and methods

### Microbial isolates and DNA extraction from colonies

Fifty-six isolates (26 *S. aureus* and 30 non-*S. aureus*) collected from the Clinical Microbiology Unit, Srinagarind Hospital, Khon Kaen University, Thailand during 2010–2019 [18] (S1 Table) were cultured on blood agar (Oxoid, Hampshire, UK) at 37°C for 24 h. DNA was extracted from colonies using the achromopeptidase method [19] or boiling methods. All fifty-six isolates were used for evaluating the performance of the chosen primer set on RPA-AGE test for detection of MRSA colonies.

## Clinical samples

Sixty blood-culture bottles, identified as positive by the BacT/Alert® Virtuo Microbial Detection System (bioMérieux, Marcy l'Etoile, France) and in which pathogens were identified by conventional biochemical tests and/or by the VITEK ®2 automated system, were obtained from Srinagarind Hospital (Table 1). All sixty samples were used for evaluating the performance of the RPA-LF test for MRSA detection (Table 1) and a subset of 30 were used to compare the efficacy of 2 brands of nucleic acid detection strips [Milenia HybriDetect vs. Kestrel Bio Sciences (KB)] (S2 Table).

The organisms other than *S. aureus* and MRSA are used for testing the specificity of the developed RPA method. A positive result must be generated only with *S. aureus* and MRSA strains, but not with the other organisms.

## Primer and probe design

A set of primers and probes was manually designed to be specific for a conserved region of the *nuc* gene following suggestions in the TwistAmp® DNA amplification assay design manual. The primers and probe for the *mecA* gene from our previous report were used [18] (Table 2).

**Table 1. Prospective evaluation of the RPA-LF assay for the detection of *nuc* and *mecA* genes in 60 positive blood-culture samples.**

| Organisms (n)[a] found in positive blood-culture samples | No. of positive samples by | | | |
|---|---|---|---|---|
| | PCR | | RPA-LF (KB)[b] | |
| | *nuc* | *mecA* | *nuc* | *mecA* |
| ***mecA*-carrying Staphylococci (16)** | | | | |
| *Staphylococcus aureus* (1) | 1 | 1 | 1 | 1 |
| coagulase-negative Staphylococci (13) | 0 | 13 | 0 | 13 |
| *Staphylococcus epidermidis* (1) | 0 | 1 | 0 | 1 |
| *Staphylococcus capitis* (1) | 0 | 1 | 0 | 1 |
| **Non-*mecA*-carrying Staphylococci (23)** | | | | |
| *S. aureus* (14) | 14 | 0 | 13 | 0 |
| coagulase-negative Staphylococci (8) | 0 | 0 | 0 | 0 |
| *Staphylococcus saprophyticus* (1) | 0 | 0 | 0 | 0 |
| **Non-*mecA*-carrying organisms (21)** | | | | |
| **Gram-positive bacteria (8)** | | | | |
| *Micrococcus* spp. (1) | 0 | 0 | 0 | 0 |
| *Bacillus* spp. (3) | 0 | 0 | 0 | 0 |
| *Enterococcus* spp. (2) | 0 | 0 | 0 | 1 |
| *Enterococcus casseliflavus* (1) | 0 | 0 | 0 | 0 |
| *Enterococcus faecalis* (1) | 0 | 0 | 0 | 0 |
| **Gram-negative bacteria (11)** | | | | |
| *Escherichia coli* (6) | 0 | 0 | 0 | 0 |
| *Klebsiella pneumoniae* (3) | 0 | 0 | 0 | 0 |
| *Salmonella* group B (1) | 0 | 0 | 0 | 0 |
| *Salmonella* group C (1) | 0 | 0 | 0 | 0 |
| **Yeast (2)** | | | | |
| *Cryptococcus neoformans* (2) | 0 | 0 | 0 | 0 |
| **Total (60)** | 15 | 16 | 14 | 17 |

[a] Pathogens were identified by conventional biochemical tests and/or by the VITEK ®2 automated system.
[b] KB, Kestrel Bio Sciences.

**Table 2. Nucleotide sequences of the primers and probes used for RPA-LF for detection of *nuc* and *mecA* genes in positive blood-cultures.**

| Primer sets | Primer names | Oligonucleotide sequences (5´ to 3´) | Length (nucleotides) | Expected products (bp) | References |
|---|---|---|---|---|---|
| nuc-set 4 | nuc-F2 | TTAAGTGCTGGCATATGTATGGCAATCGTTTC | 32 | 286 | [20] |
| | nuc-R3(RPA) | CACCATCAATCGCTTTAATTAATGTCGCAGGTTC | 34 | | This study |
| | nuc-R3-Dig-RPA-LF | Dig-CACCATCAATCGCTTTAATTAATGTCGCAGGTTC | | | This study |
| | nuc-R3-Btn-RPA-LF | Btn-CACCATCAATCGCTTTAATTAATGTCGCAGGTTC | | | This study |
| | nuc-probe | FAM-CGTAAATAGAAGTGGTTCTGAAGATCCAAC-[THF]–GTATATAGTGCAACTTC−C3-Spacer | | | This study |
| mecA-set 1 | mecA-F-(RPA_1) | GCGATAATGGTGAAGTAGAAATGACTGAACGTCCG | 35 | 176 | [21] |
| | mecA-R-(RPA_1) | TTGAACGTTGCGATCAATGTTACCGTAGTTTG | 32 | | [18] |
| | mecA-R-Btn-RPA-LF | Btn-TTGAACGTTGCGATCAATGTTACCGTAGTTTG | | | |
| | mecA-probe | FAM-CGTTAAAGATATAAACATTCAGGATCGTAA-[THF]-ATAAAAAAAGTATCTA−C3-Spacer | | | [18] |

Btn, biotin; Dig, digoxin; FAM, Carboxyfluorescein; THF, Tetrahydrofuran; C3 Spacers, a polymerase extension-blocking site.

For the TwistAmp® nfo reaction, the reverse primer was labeled with either 5'-Btn or -Dig. The probes (46 or 47 nt) include a 5´-carboxyfluorescein (FAM) antigenic label, a tetrahydrofuran (THF) spacer replacing at nt 30 and an adjacent downstream oligonucleotide (16 or 17 nt) carrying a C3-spacer (polymerase extension blocking group) at its 3´ end. Specificity of all designed RPA primers and probes was confirmed by using a BLAST search and OligoEvaluator™ Sequence Analysis software (http://www.oligoevaluator.com; last accessed April 18, 2020) [18]. Primers and probes were synthesized by Bio Basic Inc. (Makham, Ontario, Canada). The primer and probe sequences used for the RPA-LF of the *nuc* and *mecA* genes are provided in Table 2 and S3 Table. The performance of the primer set for the *nuc* gene designed in this study was compared with primer sets from previous studies by Du *et al.* [13] and Geng *et al.* [20] (S3 Table).

### Primer screening and the performance of the RPA-AGE assay for detecting *nuc* and *mecA* genes

The RPA-AGE assay using the TwistAmp® Basic kit (TwistDX, Cambridge, UK) was used to identify the best primer sets [four sets of *nuc* primers (S3 Table)]. The sensitivity and specificity of the RPA-AGE assay were evaluated for detecting *nuc* and *mecA* genes using the chosen sets of primers and genomic DNA from the panel of 56 isolates [18] (S1 Table). The total volume of a reaction was 12.5 μL. The reaction mixture included 480 nM of each RPA primer, 14 mM of magnesium acetate (MgOAc), and 0.5 μL of bacterial DNA. The tubes were incubated for 20 min at 37 and 45˚C for *nuc* and *mecA* genes respectively, and the reaction was stopped at 65˚C for 10 min. Detection of amplification products was carried out by subjecting the product to electrophoresis through a 2% agarose gel. The results of the RPA-AGE method were compared with those of the PCR-AGE approach.

### PCR assays of *nuc* and *mecA* genes for identification of MRSA

For the conventional PCR assay, the total volume used per reaction was 25 μL, including 0.5 μM of each primer of nuc-Set 4 (this study and Geng *et al.* [20]) (Table 2) or *mecA* primers

of Kondo *et al.* [22], 1X PCR buffer, 0.2 mM dNTP, 2 mM $MgCl_2$, 1 U of *Taq* DNA polymerase (Vivantis Technologies Sdn. Bhd., Selangor Darul Ehsan, Malaysia), and 2 μL of DNA template. The PCR method was carried out according to our previous study [18].

## Optimization for nuc-RPA-LF reaction

For nuc-RPA-LF, the reaction was performed at temperatures of 37, 40, 45 and 50˚C, and the optimum reaction time was determined by incubating the reaction mixtures for 5, 10, 20 and 30 min. After optimization, the nuc-RPA-LF and mecA-RPA-LF reactions were carried out separately in a total volume of 10 μL each. The master mix for RPA reaction comprised 2.1 μL of each primer [nuc-F2 & nuc-R3-Dig-RPA-LF or nuc-F2 & nuc-R3-Btn-RPA-LF [20, this study]; mecA-F-(RPA_1) & mecA-R-Btn-RPA-LF] [18, 21] (420 nM), 0.6 μL of nuc-probe or mecA-probe (120 nM), 29.5 μL of rehydration buffer and 11.2 μL of sterile distilled water. The reaction mixture was added to the freeze-dried enzyme pellet of a TwistAmp® nfo kit (TwistDx, Cambridge, UK), mixed thoroughly by pipetting, and then divided into five aliquots (9.1 μL each) into 0.2 mL tubes. The template DNA (0.4 μL) was added into each tube, and then the reaction was initiated by adding 280 mM MgOAC (0.5 μL). The reaction tube was incubated at 45˚C for 20 min. After incubation, the amplification was stopped at 82˚C for 5 min to denature the primer dimer according to the method of Liu *et al.* [23]. For LF detection, 1 μL of mecA-RPA product or nuc-RPA product using the reverse primer labeled with a Btn at the 5′ end was added into 49 μL of the HybriDetect assay buffer. The nuc-RPA product generated using the reverse primer labeled with a Dig at the 5′ end was diluted to 1:100 and 2.5 μL of the diluted product was added to a tube containing 50 μL of HybriDetect assay buffer. Finally, the HybriDetect-1 (for Btn labeled) or HybriDetect-2 (for Dig labeled) LF strip (Milenia Biotec GmbH, Gieβen, Germany) was dipped into the mixture containing the DNA product and buffer, and then left for 5 min at room temperature. The appearance of color at both the test and control lines on the strip indicates a positive result whereas a negative result shows only a control line on the strip. The absence of a control line on the LF indicates that the strip has not worked correctly.

## Performance of the Milenia HybriDetect vs. Kestrel Bio Sciences (KB) nucleic acid detection strips

Besides the Milenia strips used for detection of RPA products, DNA amplification product from 30 positive blood-culture samples (S2 Table) (subset of 60 positive blood-culture samples) were also processed using KB strips (Kestrel Bio Sciences, Pathum Thani, Thailand) following the manufacturers' instructions. Briefly, 5 μL of RPA product labeled with Btn was pipetted directly onto the sample application area of a KB strip. The strip was placed in a tube containing 50 μL of buffer and allowed to absorb up for 5 min at room temperature. Then the strips were removed and the results were inspected immediately. The interpretation of the KB strip was the same as that of the Milenia strip.

The percent of agreement and the Cohen's Kappa index value were calculated using the free software VassarStats (http://vassarstats.net/; last accessed June 26, 2021). The Kappa index value was interpreted as follows: ≤0, poor agreement; 0.01 to 0.20, slight agreement; 0.21 to 0.40, fair agreement; 0.41 to 0.60, moderate agreement; 0.61 to 0.80, substantial agreement; 0.81 to 1, almost perfect or perfect agreement [24].

## Determination of detection limit

The detection limits of RPA-AGE and RPA-LF for detection of the *nuc* and *mecA* genes were determined [18]. The KB strips were used for the detection of amplified products from both

genes. The intensity of the signal at the test line, expressed as peak area, was determined using ImageJ software (version 1.53a) (National Institute of Health, Bethesda, MD, USA) and calibration curves were generated for the detection of *nuc-* and *mecA*-carrying *S. aureus*. Above a threshold peak-area value established to be 1000, the RPA-LF was positive for detection of both *nuc* and *mecA* genes. The peak area increased with increasing cell numbers.

### Prospective evaluation of the RPA-LF test for MRSA detection directly from positive blood-culture samples

The developed nuc- and mecA-RPA-LF assays were evaluated using KB strips to test with 60 positive blood-culture samples from Srinagarind Hospital. Blood-culture samples were tested directly that no prior extraction of DNA was required. The results of the RPA-LF were compared with either conventional biochemical tests or results from the VITEK 2 system (bioMérieux) and PCR-AGE. The intensity of the signal at the test line was determined by ImageJ and interpreted as described above. Finally, the sensitivity, specificity, positive predictive value (PPV), negative predictive value (NPV) and 95% confidence interval (CI) were calculated using the free software VassarStats (http://vassarstats.net/; last accessed June 26, 2021).

All tests were performed in duplicate, and results were blindly read by two independent observers. If both results were discordant, the sample was tested for the third time and the modal result was accepted. The RPA-LF was imaged with a smartphone (Huawei Nova 2i, Huawei Base, Shenzhen, China) at a 90˚ angle and a distance of 10 cm.

This study was approved by the Ethics Committee of Khon Kaen University (project number HE611605).

## Results

### Screening of RPA primers and the performance of the RPA-AGE assay for detecting *nuc* and *mecA* genes

Our previous RPA primer set and probe for *mecA* gene detection were used in this study [18]. For the *nuc* gene, we initially evaluated four primer sets against three isolates of *S. aureus*, *S. haemolyticus*, and *E. faecium*. The sizes of DNA fragments expected to be generated by these nuc-primer sets were 139, 164, 141 and 286 bp, respectively (S3 Table). The nuc-primer set 4 showed the highest specificity with no cross-reactions. The RPA-AGE using these primer sets [nuc-set 4 & mecA-set 1, (Table 2)] provided 100% and 92.1% sensitivity for identifying *nuc* and *mecA* genes, respectively, in 56 samples. In addition, negative results were seen in all PCR-negative isolates (100% specificity) (S1 Table). These primer sets were therefore used for further validation.

### Optimization

In this study, the nuc-RPA-LF method worked well at 40–50˚C and incubation time of 10–30 min (S1 Fig). Therefore, we selected a 45˚C and 20 min as the optimal temperature and time for nuc-RPA-LF, respectively, similar to those for the mecA-RPA-LF [18].

### Performance of the Milenia HybriDetect vs. KB nucleic acid detection strips

Detection of RPA products from 30 positive blood-culture samples using the Milenia and KB strips showed a perfect agreement (100% agreement with a Cohen's Kappa index value of 1.0) (S2 Table). Btn-labeled and Dig-labeled primers gave identical results for detecting *nuc*-RPA

**Fig 1. Detection of *nuc*-RPA products by RPA-LF.** A, Milenia strips and a Btn-labeled reverse primer; B, Milenia strips and a Dig-labeled primer; C, Kestrel Bio Sciences (KB) strips and a Btn-labeled primer. 1–4, *nuc*-carrying *S. aureus*; 5, non-*nuc*-carrying *Bacillus* spp.; 6, non-*nuc*-carrying *E. coli*; 7, non-*nuc*-carrying *C. neoformans*. +, positive reaction; -, negative reaction.

products using Milenia strips (100% agreement with a Cohen's Kappa index value of 1.0) (S2 Table and Fig 1).

## Detection limit

The detection limits of the RPA-LF methods for identifying the *nuc* and *mecA* genes were 10 colony forming unit (CFU) per reaction (peak area = 1,045 and 1,809, respectively), whereas those of the RPA-AGE method for the *nuc* and *mecA* genes were 10 CFU and 1 CFU per reaction, respectively (Fig 2). With $10^4$ CFU per reaction of *mecA*-carrying *S. aureus*, the RPA-LF showed a result due to the "hook" effect [Fig 2(A-2 to C-2)].

## Prospective evaluation of the RPA-LF test for MRSA detection directly from positive blood-culture samples

The RPA-LF test provided 93.3% sensitivity, 100.0% specificity, 100.0% PPV, and 97.8% NPV for detecting the *nuc g*ene, and 100.0%, 97.7%, 94.1% and 100.0%, respectively, for the *mecA* gene (Table 1 and S4 Table). Examples of the RPA-LF results for identifying these genes in representative positive blood-culture samples and their corresponding peak areas are shown in Fig 3. The signal intensities of all positive results were higher than the threshold value (>1000 of peak area) (Fig 3). The mean values of signal intensity of *nuc*- and *mecA*-positive samples were 5857 and 4312 respectively.

## Discussion

Primer and probe design is viewed as the most challenging step for setting up an RPA-LF experiment. To avoid non-specific cross-binding, we carefully designed and tested candidate sequences for secondary-structure formation and primer-primer interactions, hairpins, nucleotide composition, sequence length, and the interplay between probe and primers. In the present study, the RPA-LF method for both genes (*nuc* and *mecA*) could be performed under the same conditions, making it convenient for routine detection.

In general, there is no single system best for all applications. We need to consider several important factors before selecting an assay including the LF strip that best fits the situation. These factors include cost, speed, ease of use, performance (acceptable sensitivity and specificity) and availability in our area. Currently, the cost of the RPA-LF assay is rather higher than other molecular methods [25]. However, a cheaper lateral-flow format or the use of alternative labeling technologies for primers may help to reduce the cost per test. The Milenia strip costs ~$4.7 per strip compared with ~$2.76 for the KB strip. In the present study, Milenia and KB strips showed perfect agreement for detecting nuc- and mecA-RPA products. In addition, the

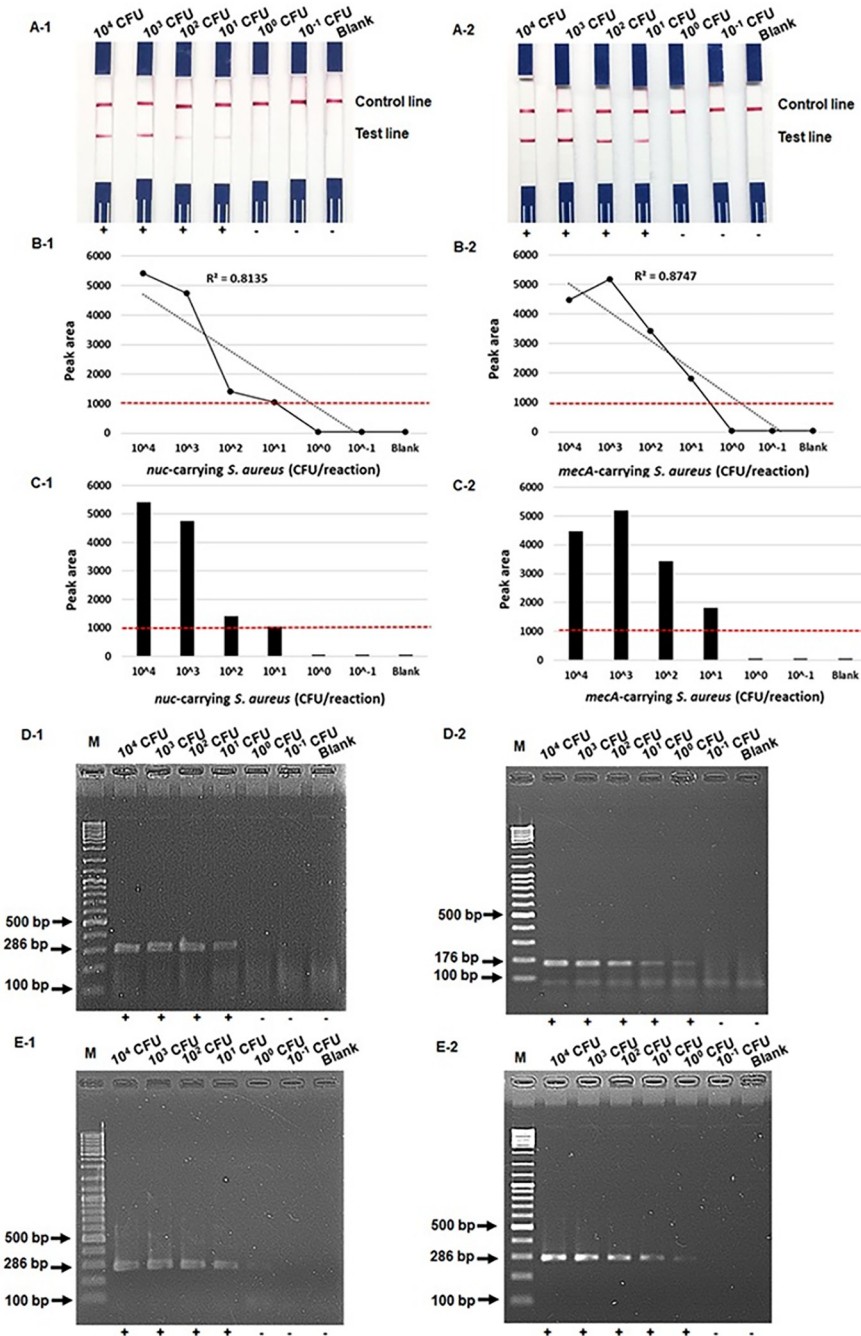

**Fig 2. Comparison of the detection limit of the RPA-LF, RPA-AGE and PCR assays.** Detection limit of the RPA-LF assay (A) compared with that of the RPA-AGE (D) and PCR (E) assays for detecting *nuc* (1) and *mecA* (2) genes. Lane M, 100 bp DNA ladder. B-1 & B-2, the calibration curve for RPA-LF with different concentrations of *nuc*-carrying *S. aureus* and *mecA*-carrying *S. aureus* corresponding to (A-1 & A-2). C-1 & C-2, Histogram representing peak area of the intensity of the test lines corresponding to (A-1 & A-2). The red dashed lines represent the threshold value (peak area = 1000) above which a result is regarded as positive.

KB strips is convenient to order and took short delivery times. The primer cost for biotin (Btn) labeling is 50% saving than that for digoxin (Dig) labeling. Therefore, we decided to use KB strips and Btn-labeled primers for the remaining experiments.

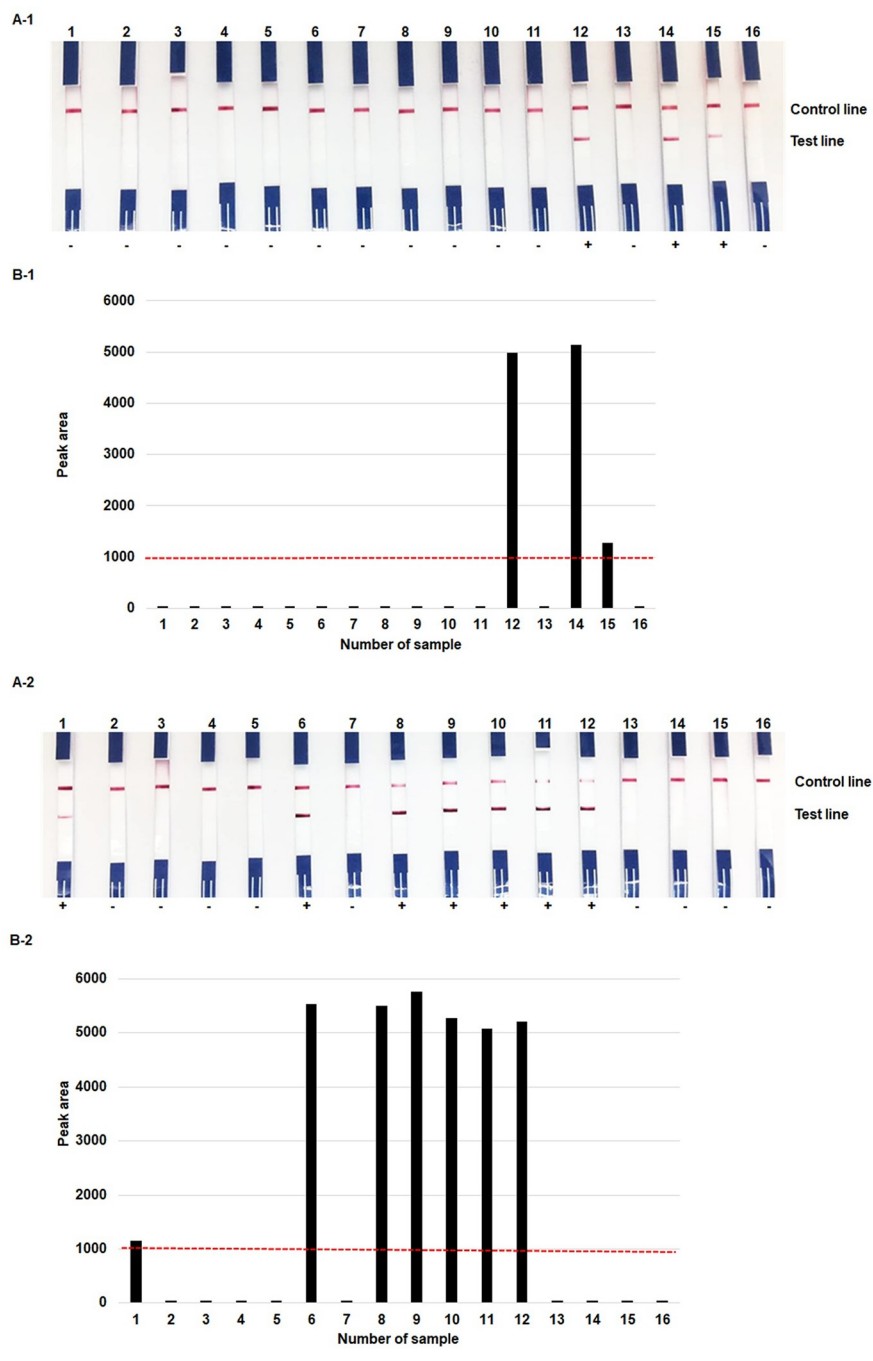

**Fig 3. Examples of RPA-LF results for the detection of *nuc* and *mecA* genes.** RPA-LF results of *nuc* (1) and *mecA* (2) genes in representative positive blood-culture samples (A) and their corresponding peak area profile plots (B). A-1 & B-1; numbers 1–11, 13 and 16, *nuc* gene-negative samples; 12, 14 and 15, *nuc* gene-positive samples. A-2 & B-2; numbers 2–5, 7, 13–16, *mecA* gene-negative samples; 1, 6, 8–12, *mecA* gene-positive samples. The threshold value of the intensity of peak area to determine a positive/negative result is 1000 (red dashed line).

In the present study, the detection limits of the RPA-LF and RPA-AGE methods for detecting the *nuc* gene were equal to that of the PCR method. For the *mecA* gene, the detection limit of RPA-LF was 10 times less sensitive than those of the RPA-AGE and PCR methods [Fig 2(A-2 to E-2)]. Although the sensitivity of the mecA-RPA-LF using KB strips was lower than that

of the RPA-AGE and PCR assays, it could nevertheless detect this gene within the concentration range that usually occurs in positive blood-culture bottles (~$10^8$–$10^9$ CFU/mL) [26] [the detection limit of the RPA-LF assay using KB strips for detecting both genes was $10^5$ CFU/mL (10 CFU per reaction)]. This test has considerable potential for routine detection of MRSA in positive blood-culture bottles.

RPA combined with LF has been shown to have high sensitivity and specificity for detecting the *nuc* gene of *S. aureus* in food [13, 27]. Recently, Brunauer *et al.* [28] reported successful direct detection of *Pseudomonas aeruginosa* by RPA-LF from wound exudate after a rapid sample preparation (crude lysis). Wang *et al.* [29] also developed a multiplex-touchdown PCR method to detect the *mecA* gene from positive blood-culture bottles. As far as we know, there is still no RPA-LF test available for the direct detection of MRSA in positive blood-culture samples. This study is the first report describing the performance of such a test evaluated using blood samples from a hospital without prior DNA extraction. However, the developed RPA-LF method could not detect the *nuc* gene of one *S. aureus* isolate in a positive blood-culture bottle (Table 1) but could detect the gene in colonies. This may be due to some inhibitor in this particular blood-culture bottle. Blood samples contain various inhibitors (e.g. lactoferrin and immunoglobulin G) that can interfere with molecular testing [30, 31]. Interference factors may hinder the enzymatic amplification reaction by direct interaction with the enzymes or by interfering with cofactors required for the enzymatic activity [30–32]. However, the high concentrations of hemoglobin seem to have no influence on a successful RPA-LF reaction [32]. In 2013, Xafranski *et al.* [33] reported that bovine serum albumin (BSA) was the most efficient protein for reducing inhibition of amplification. BSA can bind to PCR inhibitors in the samples and prevent them from interacting with DNA (*Taq*) polymerase. Therefore, it has been used to increase the sensitivity of PCR amplification from clinical samples [30, 34]. The sensitivity and the detection limit of our developed RPA-LF can be improved further.

In our previous report [18], the RPA-LF provided 100% specificity for the detection of the *mecA* gene from both colonies and spiked blood cultures. In this study, one false positive from a non-*mecA*-carrying *Enterococcus* spp. isolate in a positive blood-culture bottle from the hospital was observed. A false-positive result may be caused by any degradation products in the positive blood-culture bottle or an unknown background of the patient. However, the crowding agent (dextran sulfate) and proteins in RPA reactions can interfere and cause non-specific binding of antibody-labeled gold nanoparticles to the test line of an LF strip. Thus, the amplicon of each gene should be diluted sufficiently to avoid non-specific binding and false-positive signals [35]. Moreover, primer-primer binding at room temperature may produce a false positive signal [36]. The RPA mixtures should be kept on ice during the reaction preparation. The LF strip tended to show false positives when it was left in the running buffer for too long time [37]. We read the result of the LF test within 20 min in this study. However, the false-positive and false-negative rates of the RPA-LF in this study were ~2%.

In 2017, Yeh *et al.* [12] reported that an RPA reaction using plasma samples was much more robust than a PCR reaction or loop-mediated isothermal amplification (LAMP). The elimination of sample extraction and purification steps makes the RPA-LF method significantly less laborious, expensive and time-consuming. Of these techniques, RPA is the easiest one to perform, furthermore, its reagents in a dried pellet format are highly stable [38]. In addition, the LF strip is stable at room temperature for at least six months.

However, the RPA does have some limitations. The RPA reaction kits are produced by only one company, which has a high impact on the cost, availability and delivery times. Therefore, we tested by using reduced volume of the RPA reaction in this study (10 μL), similar to a previous report [39], which reduced the cost of RPA reagent to ~$0.5 per test. Recently, Lillis *et al.* [40] and Subbotin *et al.* [41] showed that reducing the reaction volume from 50 μL to 5 μL had

very little effect on the performance. Our study had a small sample size for the analysis because of a limited number of TwistAmp® nfo kits from the manufacturing company: larger samples should be evaluated in further studies.

In conclusion, we developed an RPA-LF test, which is simple, rapid, robust, cost-effective, highly sensitive, and specific for the direct detection of MRSA from positive blood-culture samples. This may be suitable for use in epidemiological surveillance and identification in hospitals. Moreover, it may serve as a model platform for detecting other pathogens.

## Supporting information

**S1 Fig.** Optimization of incubation temperature and time for nuc- (A, B) and mecA- (C, D) RPA-LF assays. +, positive reaction; +w, weakly positive; -, negative reaction.
(TIF)

**S1 Raw images.**
(PDF)

**S1 Table. Diagnostic performance of the RPA-AGE assay in the detection of *nuc* and *mecA* genes in 56 clinical isolate samples.**
(PDF)

**S2 Table. Performance of the RPA-LF using Milenia vs. KB strips compared with PCR assay for detecting *nuc* and *mecA* genes in positive blood-cultures from routine Srinagarind Hospital.**
(PDF)

**S3 Table. Nucleotide sequences of the RPA primers and probes tested for detection of *nuc* and *mecA* genes used in this study.**
(PDF)

**S4 Table. Diagnostic performance of the RPA-LF assay in the detection of *nuc* and *mecA* genes in 60 positive blood samples from Srinagarind Hospital.**
(PDF)

## Acknowledgments

We thank the Centre for Research and Development of Medical Diagnostic Laboratories (CMDL), Faculty of Associated Medical Sciences, Khon Kaen University, Thailand. We are grateful to Prof. Keiichi Hiramatsu, Juntendo University for providing the reference strains; Firmer Co., Ltd. for supporting pre-incubated aerobic culture bottles and Render Automated Blood Culture System. We also extend our gratitude to Tasneem Pechnur and staff of the Clinical Microbiology Unit, Srinagarind Hospital for collecting the clinical isolates. We would like to acknowledge Prof. David Blair, for editing the MS via Publication Clinic KKU, Thailand.

## Author Contributions

**Conceptualization:** Arpasiri Srisrattakarn, Aroonlug Lulitanond.

**Data curation:** Arpasiri Srisrattakarn, Aroonlug Lulitanond.

**Formal analysis:** Arpasiri Srisrattakarn, Aroonlug Lulitanond.

**Funding acquisition:** Aroonlug Lulitanond.

**Investigation:** Arpasiri Srisrattakarn, Pimchanok Panpru, Lumyai Wonglakorn.

**Methodology:** Arpasiri Srisrattakarn, Aroonlug Lulitanond.

**Project administration:** Arpasiri Srisrattakarn, Aroonlug Lulitanond.

**Writing – original draft:** Arpasiri Srisrattakarn, Aroonlug Lulitanond.

**Writing – review & editing:** Arpasiri Srisrattakarn, Pimchanok Panpru, Patcharaporn Tippayawat, Aroonwadee Chanawong, Ratree Tavichakorntrakool, Jureerut Daduang, Lumyai Wonglakorn, Aroonlug Lulitanond.

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
