## [Decision Letter · Decision Letter 0]

26 May 2022

PONE-D-22-05183Rapid detection of methicillin-resistant Staphylococcus aureus in positive blood-cultures by recombinase polymerase amplification combined with lateral flow stripPLOS ONE

Dear Dr. Lulitanond,

Thank you for submitting your manuscript to PLOS ONE. After careful consideration, we feel that it has merit but does not fully meet PLOS ONE’s publication criteria as it currently stands. Therefore, we invite you to submit a revised version of the manuscript that addresses the points raised during the review process.

We look forward to receiving your revised manuscript.

Kind regards,

Dwij Raj Bhatta, PhD

Academic Editor

PLOS ONE

Journal Requirements:

In your cover letter, please note whether your blot/gel image data are in Supporting Information or posted at a public data repository, provide the repository URL if relevant, and provide specific details as to which raw blot/gel images, if any, are not available. Email us at plosone@plos.org if you have any questions

Additional Editor Comments (if provided):

Manuscript requires corrections as pointed out but by reviewers! Please give justification: as in method section 56 isolates (26 S. aureus and 30 non-S. aureus) collected but in Table

1, there are total 60 isolates. Please check which one is correct.

2) why isolates other then S. aureus and MRSA are listed and studied? is it relevant to the title of the manuscript? study is related to S. aureus and MRSA only?

Reviewers' comments:

Reviewer's Responses to Questions

**Comments to the Author**

1. Is the manuscript technically sound, and do the data support the conclusions?

Reviewer #1: Yes

Reviewer #2: Yes

2. Has the statistical analysis been performed appropriately and rigorously? 

Reviewer #1: Yes

Reviewer #2: I Don't Know

3. Have the authors made all data underlying the findings in their manuscript fully available?

Reviewer #1: Yes

Reviewer #2: Yes

4. Is the manuscript presented in an intelligible fashion and written in standard English?

Reviewer #1: Yes

Reviewer #2: Yes

5. Review Comments to the Author

Reviewer #1: The authors described the sensitivity and specificity of RPA assay for the detection of methicillin resistant Staphylococcus aureus. The manuscript has been well written. It can accepted for publication.

Reviewer #2: The manuscript is well written and findings are important in rapid detection of MRSA from blood culture. However, some changes are suggested

1) In Material and method section it is mentioned that, 56 isolates (26 S. aureus and 30 non-S. aureus) collected but in Table

1, there are total 60 isolates. Please check which one is correct.

2) Organism other than S. aureus and MRSA are listed. As per the title of the manuscript, study is related to S. aureus and MRSA only.

6. PLOS authors have the option to publish the peer review history of their article (what does this mean?). If published, this will include your full peer review and any attached files.

Reviewer #1: **Yes: **Megha Raj Banjara

Reviewer #2: **Yes: **Dharm Raj Bhatta

---

## [Author Response · Author response to Decision Letter 0]

14 Jun 2022

June 1, 2022

Dear Editor,

Manuscript ID: PONE-D-22-05183

Title: Rapid detection of methicillin-resistant Staphylococcus aureus in positive blood-cultures by recombinase polymerase amplification combined with lateral flow strip

Thank you very much for valuable comments and for giving us an opportunity to revise our manuscript. We have responses to all the issues raised by the editor and reviewers as in the following and use the "Track Changes" function in Microsoft Word of the revised manuscript. We would be grateful if you would consider our revised manuscript for publication. 

Yours sincerely,

Aroonlug Lulitanond

Corresponding author

The response to editor’s and reviewer’s comments

Journal Requirements:

Response: We have already checked PLOS ONE style as suggested. Our manuscript was prepared accordingly the PLOS ONE style.

In your cover letter, please note whether your blot/gel image data are in Supporting Information or posted at a public data repository, provide the repository URL if relevant, and provide specific details as to which raw blot/gel images, if any, are not available. Email us at plosone@plos.org if you have any questions

Response: Our manuscript provided the original images for all gel data reported in our submission. We already checked the policy and the journal’s other requirements for blot/gel reporting and figure preparation as suggested. We have confirmed that our figures adhere to the guidelines. Our blot/gel image data are not in Supporting Information or posted at a public data repository.

Response: The phrases “data not shown” in our manuscript were removed as suggested. The data are not a core part of the research. We have already checked carefully reference list and confirmed that it is complete and correct.

Additional Editor Comments (if provided):

1. Manuscript requires corrections as pointed out but by reviewers! Please give justification: as in method section 56 isolates (26 S. aureus and 30 non-S. aureus) collected but in Table

1, there are total 60 isolates. Please check which one is correct.

Response: We have already checked as suggested. The 56 isolates (26 S. aureus and 30 non-S. aureus) were used for evaluating the performance of the chosen primer set on the RPA-AGE test for detection of MRSA colonies (S1 Table) whereas the 60 samples were clinical blood-culture bottles from a routine laboratory in hospital which were used for evaluating the performance of RPA-LF test (Table 1). So, we have specified this point in lines 86-88, 90-94, and 126-128.

2) why isolates other than S. aureus and MRSA are listed and studied? is it relevant to the title of the manuscript? study is related to S. aureus and MRSA only?

 Response: The title of the manuscript is related to S. aureus and MRSA only because we want to develop the RPA assay for the detection of nuc and mecA genes to diagnose S. aureus and its methicillin-resistance status. The organisms other than S. aureus and MRSA are included for testing the specificity of the developed RPA method. A positive result must be generated only with S. aureus and MRSA strains, but not with the other organisms. We have added this matter in lines 98-99.

Reviewers' comments:

Responses to Reviewer's Questions

Comments to the Author

1. Is the manuscript technically sound, and do the data support the conclusions?

Reviewer #1: Yes

Reviewer #2: Yes

Response: Thank you very much.

2. Has the statistical analysis been performed appropriately and rigorously?

Reviewer #1: Yes

Reviewer #2: I Don't Know

Response: Thank you very much. We have described the sensitivity and specificity (statistical analysis) of RPA assay for the detection of methicillin resistant Staphylococcus aureus.

3. Have the authors made all data underlying the findings in their manuscript fully available?

Reviewer #1: Yes

Reviewer #2: Yes

Response: Thank you very much.

4. Is the manuscript presented in an intelligible fashion and written in standard English?

Reviewer #1: Yes

Reviewer #2: Yes

Response: Thank you very much.

5. Review Comments to the Author

Reviewer #1: The authors described the sensitivity and specificity of RPA assay for the detection of methicillin resistant Staphylococcus aureus. The manuscript has been well written. It can accepted for publication.

Response: Thank you very much for accepting our manuscript for publication.

Reviewer #2: The manuscript is well written and findings are important in rapid detection of MRSA from blood culture. However, some changes are suggested

1) In Material and method section it is mentioned that, 56 isolates (26 S. aureus and 30 non-S. aureus) collected but in Table 1, there are total 60 isolates. Please check which one is correct.

 Response: We have already checked as suggested. The 56 isolates (26 S. aureus and 30 non-S. aureus) were used for evaluating the performance of the chosen primer set on the RPA-AGE test for detection of MRSA colonies (S1 Table) whereas the 60 samples were clinical blood-culture bottles from a routine laboratory in hospital which were used for evaluating the performance of RPA-LF test (Table 1). So, we have specified this point in lines 86-88, 90-94, and 126-128.

2) Organism other than S. aureus and MRSA are listed. As per the title of the manuscript, study is related to S. aureus and MRSA only.

Response: The title of the manuscript is related to S. aureus and MRSA only because we would like to develop the RPA assay for the detection of nuc and mecA genes to diagnose S. aureus and its methicillin-resistance status. The organisms other than S. aureus and MRSA are included for testing the specificity of the developed RPA method. A positive result must be generated only with S. aureus and MRSA strains, but not with the other organisms. We have added this matter in lines 98-99.________________________________________

6. PLOS authors have the option to publish the peer review history of their article (what does this mean?). If published, this will include your full peer review and any attached files.

Do you want your identity to be public for this peer review? For information about this choice, including consent withdrawal, please see our Privacy Policy.

Reviewer #1: Yes: Megha Raj Banjara

Reviewer #2: Yes: Dharm Raj Bhatta

Response: Thank you very much.

---

## [Editor Report · Decision Letter 1]

16 Jun 2022

Rapid detection of methicillin-resistant Staphylococcus aureus in positive blood-cultures by recombinase polymerase amplification combined with lateral flow strip

PONE-D-22-05183R1

Dear Dr. Lulitanond,

We’re pleased to inform you that your manuscript has been judged scientifically suitable for publication and will be formally accepted for publication once it meets all outstanding technical requirements.

Kind regards,

Dwij Raj Bhatta, PhD

Academic Editor

PLOS ONE
---

## [Editor Report · Acceptance letter]

22 Jun 2022

PONE-D-22-05183R1 

Rapid detection of methicillin-resistant *Staphylococcus aureus* in positive blood-cultures by recombinase polymerase amplification combined with lateral flow strip 

Dear Dr. Lulitanond:

I'm pleased to inform you that your manuscript has been deemed suitable for publication in PLOS ONE. Congratulations! Your manuscript is now with our production department. 

Kind regards, 

on behalf of

Professor Dwij Raj Bhatta 

Academic Editor

PLOS ONE